# A Water-Stable Zn-MOF Used as Multiresponsive Luminescent Probe for Sensing Fe^3+^/Cu^2+^, Trinitrophenol and Colchicine in Aqueous Medium

**DOI:** 10.3390/ma15197006

**Published:** 2022-10-09

**Authors:** Xiaojing Zhou, Lili Liu, Yue Niu, Mingjun Song, Yimin Feng, Jitao Lu, Xishi Tai

**Affiliations:** School of Chemical & Chemical Engineering and Environmental Engineering, Weifang University, Weifang 261061, China

**Keywords:** MOFs, luminescent sensing, Fe^3+^/Cu^2+^, trinitrophenol, colchicine

## Abstract

A water-stable Zn-MOF was constructed based on H_2_PBA and 1, 10-phenanthroline under solvothermal conditions. The compound exhibited a 3D (2,3,8)-connected (4^3^)_2_(4^6^.6^6^.8^15^.12)(8) topology framework. The crystal structure and phase purity of the compound was verified by single crystal X-ray diffraction. Subsequently, some studies on the morphology, structure, and luminescent properties were carried out. The results showed that this compound could be used as a versatile chemosensor for Fe^3+^/Cu^2+^, trinitrophenol and colchicine via a luminescence quenching effect in an aqueous medium.

## 1. Introduction

In the past decades, the rapid growth of industry and population has increased the demand for clean water everywhere, which has therefore become a vital area of research [1,2,3]. Water pollution including heavy metals ions, anions, organics, antibiotic, dyes and explosives are harmful to human health and the ecological environment. Therefore, considerable attention should be focused on the detection and assessment of water pollution by sensitive and reliable sensors.

Fe^3+^ and Cu^2+^ are two well-known important ions, in which Fe^3+^ is essential for the human body; excess or the deficiency of Fe^3+^ results in various health problems such as Alzheimer’s disease, hemochromatosis, diabetes, etc. [4,5]; meanwhile, Cu^2+^ plays an important role in all living organisms and is vital to their health; however, an excessive intake of Cu^2+^ will cause an imbalance in the cellular processes and leads to increased blood pressure, and damage to the kidneys and liver [6]. As a consequence, it is highly desirable to detect Fe^3+^ and Cu^2+^ for human health. Meanwhile, colchicine has been discovered to inhibit many inflammatory processes, it is a toxic natural product which is originally extracted from plants of the genus Colchicum; even trace amount of colchicine may cause emesis, kidney failure, diarrhea, and bone marrow suppression to human beings, colchicine is also a classical drug for treating gout, spontaneous inflammation, liver cirrhosis [7,8].

On the other hand, nitroaromatics are highly toxic and strongly explosive; the detection of these explosive nitroaromatics is of great significance to human security, anti-terrorism handling and environmental protection. Among the nitroaromatics, as a kind of vital chemical raw material, trinitrophenol (TNP) has been extensively used in the area of explosives, papermaking, medicines, and textiles. However, the discharge of trinitrophenol into the environment may cause male infertility, anemia, and other severe health problems. Therefore, it is urgent to find an effective and rapid technique to detect these pollutants [9,10,11,12,13].

Up to now, numerous monitoring and detection techniques have been employed for sensing metal ions, trinitrophenol or colchicine in recent years, such as atomic absorption apectroscopy, raman spectroscopy, inductively coupled plasma-mass spectrometry, ion mobility spectrometry, electrochemical methods, etc. [14,15]. These detecting procedures are time consuming, requiring sophisticated sample preparation methods. In comparison, the luminescence sensor has recently become very efficient in terms of its convenient visual detection. Meanwhile, fewer reports involving the fluorescence detection of colchicine. Hence, it may help to pave the way for fluorescence detection of toxic contaminants in water systems [16,17].

Metal organic frameworks (MOFs) have been widely studied owing to their potential applications in various fields such as gas storage and separation, food safety, sensing, catalysis and drug delivery [18,19,20,21,22,23,24]. Among them, luminescent MOFs (LMOFs) have been widely investigated in the area of chemical sensing due to their outstanding luminescent properties. For instance, Li et al. reported the first high LMOF for detecting trace amounts of nitroaromatics in the vapor phase; Zhao and co-workers synthesized an anionic In-MOF which is the first example of a MOF-based luminescent probe to detect colchicine; Li and Zhang et al. constructed a robust Tb(III)-MOF for the highly sensitive sensing of Cu^2+^ and nitromethane. These results show that MOFs can act as effective fluorescent probes for sensing metal ions, nitroaromatics or colchicine [25,26].

At the same time, most of the LMOFs were exploited in organic solvent systems but not in a water system, because of their poor water stability, which limits their practical application. Up to now, only a few of the MOF-based sensors reported can maintain their stability in water. Therefore, there is an urgent need to develop new chemically stable LMOFs for the selective and fast detection of metal ions, nitroaromatics or colchicine in water systems [27].

In this work, we chose Zn^2+^ as the metal source as the d10 transition metal ion, as it is useful to construct LMOFs with unique photophysical properties [28,29,30,31].

Based on the above consideration, in this study, we present a fluorescent Zn-MOF, Zn_4_(PBA)_3_(OH)_2_(DMF) under solvothermal conditions which can be utilized as multifunctional material for sensing Fe^3+^/Cu^2+^, trinitrophenol and colchicine in aqueous medium with high sensitivity. The crystal structure was characterized by IR spectroscopy, thermogravimetric, single-crystal, powder X-ray diffraction methods. Furthermore, the luminescent property and sensing behavior of Fe^3+^/Cu^2+^, trinitrophenol, colchicine for the Zn-MOF in water was also investigated in detail. It is shown that the Zn-MOF can be used as effective fluorescence sensor for sensing Fe^3+^/Cu^2+^, trinitrophenol and colchicine. Moreover, the quenching mechanism is also discussed briefly.

## 2. Physical Measurements

FT-IR spectra were collected on a Nicolet impact 410 FTIR spectrometer from USA nicolet corporation with KBr pellet in the 400–4000 cm^−1^ region. Solvothermal reaction was performed in 15 mL Teflon-lined stainless steel autoclave.

Elemental analyses for C, H, N were performed on a Perkin-Elmer 2400 element analyzer from USA Perkin-Elmer corporation. Powder X-ray diffraction measurement data were collected on a Rigaku D/max 2550 X-ray Powder Diffractometer from Japan Rigaku corporation (Table 1). TGA was performed using a TGA Q500 V20.10 Build 36 from USA Ta instruments corporation, which was collected from room temperature to 800 °C at a heating rate of 10 °C/min in a flowing N_2_ atmosphere. UV-Vis spectra were acquired from a TU-1900 spectrometer from Beijing puxi general instruments corporation at room temperature. Luminescent spectra were obtained on the FLS920 spectrofluorometer from Edinburgh company of UK, both the excitation and emission pass width were 2.0 nm.

## 3. Synthesis of the Compound

Zinc(II) nitrate hexahydrate (29.7 mg from Sinopharm, 99.99%), 3,5-Pyridinedicarboxylic acid (8.4 mg), and 1,10-phenanthroline (15.6 mg), DMF (5 mL) were mixed in a 15 mL autoclave and kept in an oven at 100 °C for 72 h; this was then allowed to cool to room temperature, octahedral crystals were collected and washed with DMF, followed by drying at 60 °C. Yield 58% (based on 3,5-Pyridinedicarboxylic acid), IR (KBr 4000–400 cm^−1^) 3435 (w), 3055 (w), 1643 (s), 1614 (s), 1522 (m), 1434 (m), 1280 (m), 1146 (w), 1029 (w), 846 (m), 767 (m), 737 (m), 641 (w), 587 (l). Elemental analysis (%): Calcd for: C24H18N4O16Zn4: C 32.7 H 2.0 N 7.73; found: C 32.53 H 1.96 N 7.65.

**Table 1 materials-15-07006-t001:** Crystallographic data and structure refinement summary for the compound.

Compound	Zn-MOF
Molecular formula	C24H18N4O16Zn4
Formula weight	879.82
Crystal system	Monoclinic
Space group	*C 2/c*
*a*, Å	26.7137(7)
*Β*, Å	11.1860(3)
*c*, Å	19.7493(5)
*α*	90
*β*	132.56
*γ*	90
*V*, Å^3^	4347.1(2)
*Z*	4
*D_calc_*, g/cm^3^	1.260
*F*(000)	1632.0
*GoF*	1.057
*R*_1_, *wR*_2_ [I > 2σ(I)] ^a,b^	*R*_1_ = 0.036, *wR*_2_ = 0.1029
*R*_1_, *wR*_2_ (all data)	*R*_1_ = 0.0435 *wR*_2_ = 0.1059

^a^*R*_1_ = ||*F*_o_| − |*F*_c_||/Σ|*F*|_o_. ^b^ wR_2_ = [Σ*w*(*F*_o_^2^ − *F*_c_^2^)^2^/Σ*w*(*F*_o_^2^)^2^]^1/2^. *w* = 1/ [σ^2^(*F*_o_^2^) + (*ap*)^2^ + (*bp*)], *p* = [max(*F*_o_^2^ or 0) + 2(*F*_c_^2^)]/3. a = 0.0540, b = 19.106.

## 4. Crystal Structure of the Compound

Crystallographic study revealed that the compound crystallized in space group C2/c and its asymmetric unit contained two Zn centers, one and a half 3, 5-Pyridinedicarboxylate groups, one hydroxide(OH^−^) anion and one coordinated water molecule (Appendix A). There were two kinds of Zn coordination modes, Zn1 was hexacoordinated by one hydroxide anion, three oxygen atoms from three PBA^2−^ ligands, and one nitrogen atom from one PBA^2−^ ligand, one water molecule, Zn2 adopted a pentagonal geometry, coordinated by three oxygen atoms from three different PBA^2−^ molecules (Figure 1), the Zn-O distances were from 1.928(13) to 2.151(4) Å, the Zn-N bond length was 2.170(3) Å (Appendix A). The PBA^2−^ ligands showed two different coordination modes (L_a_ and L_b_) (Appendix A). Each carboxylate group adopted µ_2_-η^1^:η^1^ bidentate mode, the hydroxide anion showed a µ_2_- bridging mode to coordinate to one Zn(1) and one Zn(2) center. To describe the extended framework of the compound, it was noted that two Zn(1) and Zn(2) centers are held together by two µ_2_-OH and six carboxylate groups from six PBA^2−^ ligands to form a tetranuclear zinc cluster, {Zn_4_(OH)_2_(CO_2_)_6_}, which was further connected by eight PBA^2−^ ligands to construct a 3D framework (Figure 2). From the topological viewpoint, L_a_ could be considered as a 3-connected node, while L_b_ could be defined as a linker and the [Zn_4_(COO)_8_] cluster could be viewed as an 8-connected node. Thus, the overall structure of this compound was a 3D (2,3,8)-connected framework with (4^3^)_2_(4^6^.6^6^.8^15^.12)(8) topology (Figure 3). Topological classification with the TOPOS TTD collection revealed that such a net has never been found in coordination polymers. The solvent accessible volume of the compound was estimated to be 48.7% of the total unit cell volume by using PLATON software.

## 5. Luminescent Emission

The solid state fluorescent properties of the compound and H_2_PBA were investigated, as shown in Appendix A. H_2_PBA (λ_ex_ = 320 nm) obtained the maximum emission peak at 377 nm, which can be attributed to the π*-π transitions of the conjugate structure, and the compound obtained the maximum emission peaks at 370 nm, 387 nm (λ_ex_ = 320 nm), which can be attributed to MLCT or LMCT process.

The fluorescent response of the compound towards various solvents was investigated to verify their luminescent character, DMF, DMA, DMSO, NMP (N-Methyl-2-pyrrolidone), ethanol, acetonitrile, methanol, ethyl acetate, ethylene glycol and H_2_O were selected in this work. Basically, 3 mg of the compound was well dispersed in 3 mL solvent and ultrasonicated for 30 min (3 mg/3 mL), leading to stable suspensions. As shown in Figure 4, the fluorescence intensities of the compound were highly depended on the solvents (λ_ex_ = 320 nm); notably, in H_2_O, the compound demonstrated the strongest fluorescence emission, thus, H_2_O was chosen as the dispersed solvent to find the potential fluorescence information.

To explore whether the compound could detect metal ions, 3 mg samples of the compound were sonicated in 3 mL M(NO_3_)_x_ water solution (0.01 M) (M = Al^3+^, Cd^2+^, Cr^3+^, Cu^2+^, Fe^3+^, K^+^, Mn^2+^, Na^+^, Pb^2+^, Zn^2+^) for 30 min to give uniform suspensions (3 mg/3 mL). The luminescence intensities of the suspensions were measured, as illustrated in Figure 5, the luminescence intensities were heavily dependent on the metal ions (λ_ex_ = 320 nm). Among them, Fe^3+^ and Cu^2+^ showed the most significant quenching effect. To explore the water stability of the compound, the as-synthesized sample was immersed in distilled water for 24 h. As illustrated in Appendix A, the PXRD pattern of this compound still remained stable. Meanwhile, the PXRD of the compound was measured after sensing the metal ions which retained their structural integrity (Appendix A).

In order to evaluate the sensing sensitivity of the compound towards Fe^3+^/Cu^2+^, 5 × 10^−3^ M Fe^3+^ or Cu^2+^ was gradually added to the compound (3 mg) in 3 mL distilled water (3 mg/3 mL) for fluorescence titration experiments (λ_ex_ = 320 nm). With the increased volume of Fe^3+^/Cu^2+^ for the compound, the fluorescence intensity decreased continuously (Figure 6 and Figure 7). This quenching effect was analyzed by linear Stern-Volmer (S-V) equation: *I_0_*_/_*I* = 1 + *K*_sv_ [M], where *K*_sv_ and [M] refer to the quenching constant and the concentration of metal ions, *I*_0_ represents the initial fluorescence intensity, and *I* denotes the intensities after the addition of Fe^3+^ or Cu^2+^. The Stern-Volmer plots for Fe^3+^ or Cu^2+^ were nearly linear with a correlation coefficient of 0.95, 0.97 for the compound, and the slope of *K*_sv_ was calculated to be 3.64 × 10^4^ M^−1^, 6.83 × 10^3^ M^−1^ (Appendix A), which was higher than some reported results [32,33,34]. According to the equation of LOD = 3σ/k, the limits of detection of Fe^3+^ and Cu^2+^ were 1.95 mM, 10.4 mM respectively, and the results indicated that the compound could be use as sensitive luminescence probe toward Fe^3+^/Cu^2+^. The presence of Fe^3+^/Cu^2+^ aqueous solution almost completely quenched the luminescence intensity of the compound for the given metal ions (Appendix A).

To trace their potential for nitroaromatic detection, the fluorescence characteristics for the compound were studied. The nitroaromatics, used in this work included 2,4-nitrotoluene (2,4-NT), 2-nitrotoluene (2-NT), trinitrophenol (TNP), 2,4,6-trinitrotoluene (TNT), 4-nitrotoluene (p-NT), nitrotoluene (NT). Upon incremental addition of the nitroaromatic aqueous solutions (1 mM), gradual quenching of the fluorescence intensities at 383 nm of the compound were observed (λ_ex_ = 320 nm), whereas very remarkable levels of fluorescence quenching were observed through the gradual addition of trinitrophenol aqueous solution (Figure 8).

To further investigate the sensitivity of the compound for the detection of trinitrophenol, the fluorescence-quenching titration study was performed by the gradual addition of trinitrophenol aqueous solution (5 × 10^−4^ M) to the suspensions of the compound; the results showed that the luminescent intensity drastically decreased (Figure 9). According to the titration data, the Stern-Volmer plots were observed using the *I*_0_/*I* = 1 + *K*_sv_ [M] equation, a nearly linear correlation between the quenching rate and the concentration of trinitrophenol could be obtained, meanwhile *K*_sv_ for trinitrophenol was calculated 1.97 × 10^5^ M^−1^ (*R*^2^ = 0.9848) (Appendix A), which is higher than other reported MOF sensors [35,36,37], and the limit detection of trinitrophenol is 0.35 mM.

The fluorescence response of the compound in colchicine aqueous solutions (1 × 10^−2^ M) was also examined (λ_ex_ = 320 nm) (Appendix A); the fluorescence intensities were decreased dramatically, suggesting high sensitivity of the compound for colchicine. Subsequently, a fluorescence titration experiment for colchicine was performed in which the suspensions of the compound was added with an incremental volume of colchicine aqueous solutions (50–1000 μL, 5 × 10^−4^ M). it was observed that the luminescence intensity of the compound gradually decreased (Figure 10). Stern-Volmer plots were acquired according to the titration data; the results showed a higher *K*_sv_ value of 1.05 × 10^5^ M^−1^ to detect colchicine (*R*^2^ = 0.975) (Appendix A) [38], the limit of detection of colchicine was 0.66 mM. These results reveal that the compound has proper sensitivity to detect colchicine in aqueous medium.

In addition, the Fe^3+^/Cu^2+^_,_ trinitrophenol and colchicine aqueous solutions exhibited an absorption in the 260–500 nm range, which had overlaps with the excitation of the compound (Appendix A). This shows the energy of excited light is taken by Fe^3+^/Cu^2+^, trinitrophenol and colchicine, so the transfer energy of the compound is blocked, resulting in the quenching effect. The sensing mechanism for metal ions can be attributed to the suppression of luminescence resonance energy transfer and the enhancement of intermolecular electron transfer [39].

## 6. Conclusions

In summary, a water-stable fluorescent Zn-MOF was constructed using a solvothermal method, the compound feathers a 3D (2,3,8)-connected framework with (4^3^)_2_(4^6^.6^6^.8^15^.12)(8) topology. Luminescence–sensing investigations showed that the compound could be developed as a multifunctional material for high detection of Fe^3+^/Cu^2+^, trinitrophenol and colchicine. It is highly anticipated that this versatile detection performance may shed light on the design of more multifunctional MOFs in chemical sensing.

## Figures and Tables

**Figure 1 materials-15-07006-f001:**
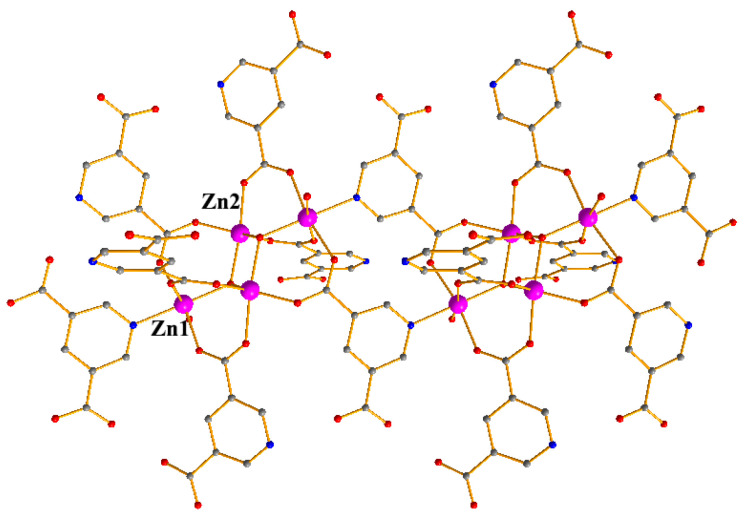
The coordination of Zn ions in the compound.

**Figure 2 materials-15-07006-f002:**
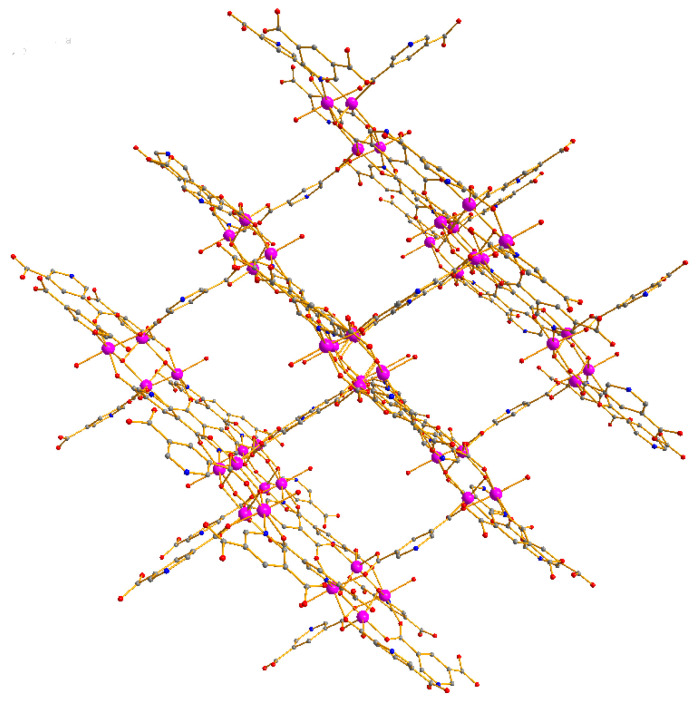
The 3D structure of the compound.

**Figure 3 materials-15-07006-f003:**
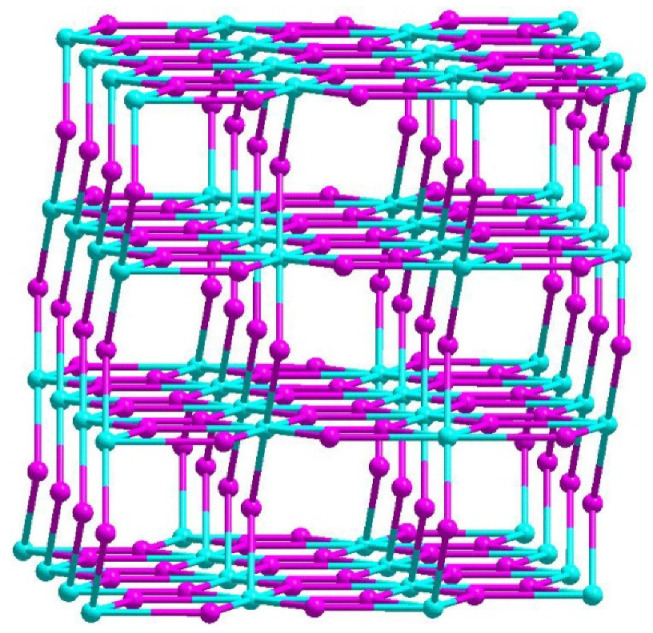
The topology of the compound.

**Figure 4 materials-15-07006-f004:**
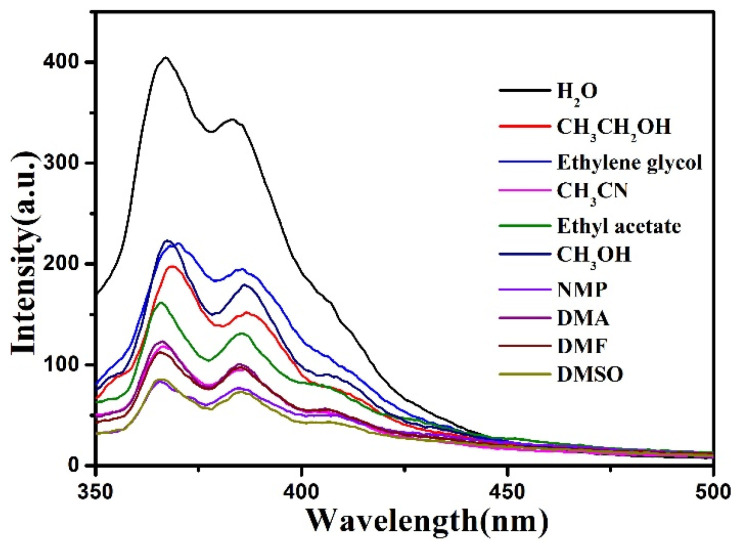
Fluorescence spectra of the compound in different solvents.

**Figure 5 materials-15-07006-f005:**
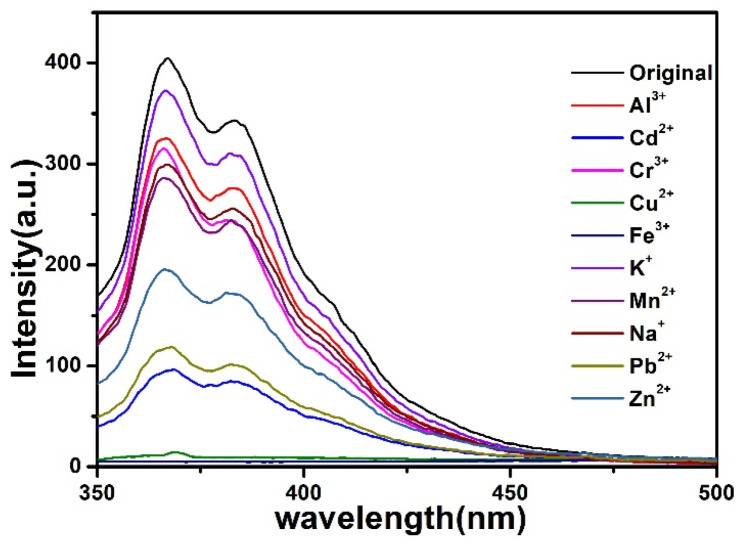
Fluorescence spectra of the compound towards different metal ions.

**Figure 6 materials-15-07006-f006:**
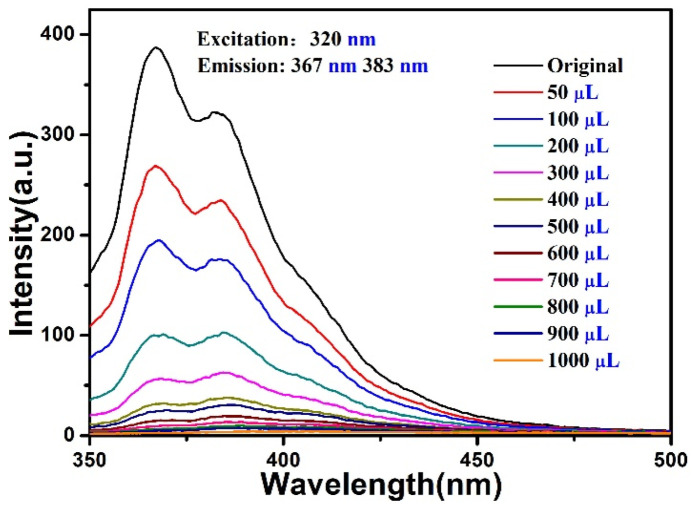
Fluorescence spectra of the compound with addition of Fe^3+^ (5 × 10^−3^ M).

**Figure 7 materials-15-07006-f007:**
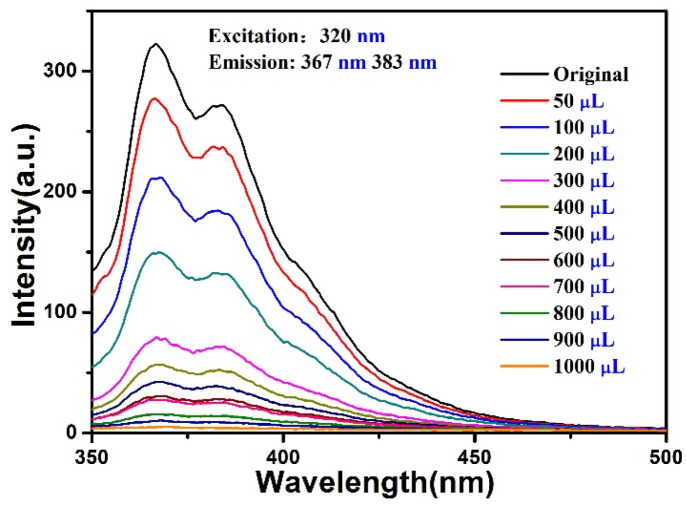
Fluorescence spectra of the compound with addition of Cu^2+^ (5 × 10^−3^ M).

**Figure 8 materials-15-07006-f008:**
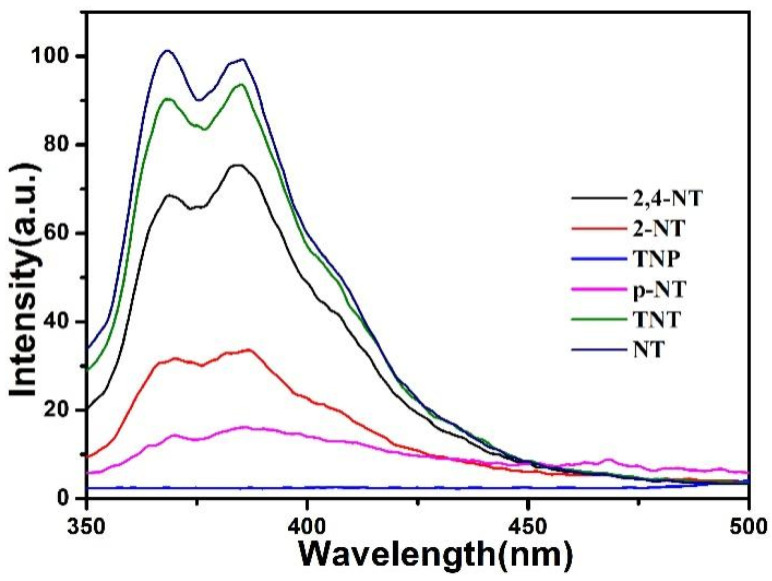
Fluorescence spectra of the compound towards explosives (1 × 10^−3^ M).

**Figure 9 materials-15-07006-f009:**
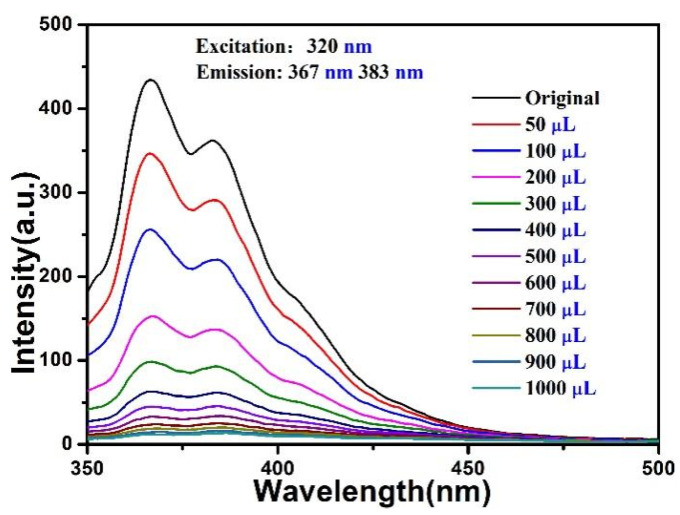
Fluorescence spectra of the compound towards trinitrophenol (5 × 10^−4^ M).

**Figure 10 materials-15-07006-f010:**
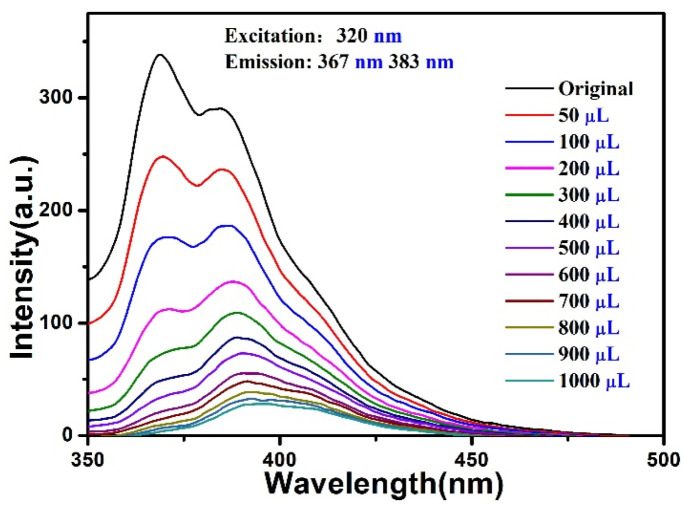
Fluorescence spectra of the compound towards colchicine (5 × 10^−4^ M).

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
