# Peer review of "A Water-Stable Zn-MOF Used as Multiresponsive Luminescent Probe for Sensing Fe3+/Cu2+, Trinitrophenol and Colchicine in Aqueous Medium"

_materials, 2022, doi:10.3390/ma15197006_

Round 1
Reviewer 1 Report
The article submitted by Lu, Tai, and co-workers reports the design, synthesis, and characterization of water-stable photoluminescent Zn-MOF for sensing of multiple analytes such as Fe3+/Cu2+, 2,4,6-trinitrophenol (TNP) and colchicine. The authors have used techniques such as FT-IR, CHN, single crystal XRD, TGA, and emission spectroscopy to characterize the reported MOF. The sensing studies were conducted by monitoring the emission changes of the MOF in the presence of various analytes studied. Overall the work is well executed, and the experimental results support the objective of the work. I recommend publishing this work after minor revisions in the Materials journal.
Specific points:
- It is unclear what is the driving force for the specific interaction with reported MOF with TNP and colchicine.
- The authors emphasize the synthesized MOFs are water stable – however, adequate experimental results are missing.
- Provide the expansion of all abbreviations used in the paper (for instance – abbreviations of nitroaromatics used in work)
- Physical measurement: The details of steady-state emission spectroscopy appear in two places. “Both the excitation and emission pass width are 2.0 nm. Luminescent spectras were obtained on the FLS920 Spectro fluorimeter.
- It could be helpful if the authors provide the concentration of MOF dispersion and the concentration of various analytes used in the figure caption. Similarly, give the excitation wavelength.
- Correct/modify the following: Tittle: Fe3+/Cu2+; replace spectras with spectra; use SV for Stern-Vomer plot.
- It could be nice if authors reduce the grammatical mistakes in the main text.

Author Response
The article submitted by Lu, Tai, and co-workers reports the design, synthesis, and characterization of water-stable photoluminescent Zn-MOF for sensing of multiple analytes such as Fe3+/Cu2+, 2,4,6-trinitrophenol (TNP) and colchicine. The authors have used techniques such as FT-IR, CHN, single crystal XRD, TGA, and emission spectroscopy to characterize the reported MOF. The sensing studies were conducted by monitoring the emission changes of the MOF in the presence of various analytes studied. Overall the work is well executed, and the experimental results support the objective of the work. I recommend publishing this work after minor revisions in the Materials journal.
Specific points:
1:It is unclear what is the driving force for the specific interaction with reported MOF with TNP and colchicine.
Response: 1. Thank you for your suggestions,we have done the UV adsorption of MOF, TNP and colchicine, we put the graph in Fig. S13, and explained the interaction of them at the end of this manuscript, then we marked it blue.
2:The authors emphasize the synthesized MOFs are water stable, however, adequate experimental results are missing.
Response: 2. Thank you for your suggestions,we have measured the PXRD of the compound after immersing in distilled water for 24 h, the structure of the compound remained stable, and we put the corresponding graph in the supporting information (Figure. S4).
3:Provide the expansion of all abbreviations used in the paper (for instance – abbreviations of nitroaromatics used in work)
Response: 3. Thank you very much, according to your suggestions, we have provided the expansion of the abbreviations in this manuscript, and marked them in blue.
4:Physical measurement: The details of steady-state emission spectroscopy appear in two places. “Both the excitation and emission pass width are 2.0 nm. Luminescent spectras were obtained on the FLS920 Spectro fluorimeter.
Response: 4. Thank you for your suggestions, we have put them together, and marked them in blue in the manuscript.
5:It could be helpful if the authors provide the concentration of MOF dispersion and the concentration of various analytes used in the figure caption. Similarly, give the excitation wavelength.
Response: 5. Thank you very much for your suggestions, we have provided the concentration of MOF and analytes in the manuscript, we also added the excitation wavelength in the paper.
6:Correct/modify the following: Tittle: Fe3+/Cu2+; replace spectras with spectra; use SV for Stern-Vomer plot.
Response: 6. Thank you for your suggestions, we have modified the above problems, and marked them in blue in the manuscript.
7:It could be nice if authors reduce the grammatical mistakes in the main text.
Response: 7. Thank you for your suggestions, we have checked the manuscript carefully, we revised the wrong words, and modified some grammatical mistakes, then marked them in blue.
Reviewer 2 Report
This work is devoted to the syntesis and structural characterization of new Zn(II)-based MOF and further luminescent detection study using this MOF as a sensor. Thousands of works report luminescent sensing of Fe3+ and nitroaromatic compounds, and the detection limits presented here are quite high, so this part of the manuscript does not possess a significant novelty. On the contrary, authors additionally report interesting results on the detection of Cu2+ and colchicine, which improve considerably the overall impression of the work. In general, the paper can be accepted for publication after performing some significant corrections:
1. ESI file is now absent in the submission, so the mentioned PXRD data are uncheckable. Please add ESI to the revised submission.
2. PXRD confirmation of the framework stability in H2O is not mentioned. Please add this information to the manuscript or ESI.
3. Fe3+, Cu2+, NACs and colchicine have their own strong absorption in UV, so the UV absorption of free moietties in solutions can easily affect the observable concentrational quenching. To confirm the real impact of these species on MOF luminescence, their incorporation needs to be additinally proven by EDX/OES (for metals) or by CHN/IR/NMR/another method (for organics).
4. line 11 - the compound is not based on phenanthroline, despite using this reagent in the synthesis. Please correct this sentence.
5. At least some assumptions concerning the mechanism of metal ion / NAC / colchicine impact on the MOF luminescence should be suggested in the main text.
6. Of course most chemists know main applications of colchicine, but it would be fair to add a brief description of colchicine importance into the introduction part.
Author Response
This work is devoted to the syntesis and structural characterization of new Zn(II)-based MOF and further luminescent detection study using this MOF as a sensor. Thousands of works report luminescent sensing of Fe3+ and nitroaromatic compounds, and the detection limits presented here are quite high, so this part of the manuscript does not possess a significant novelty. On the contrary, authors additionally report interesting results on the detection of Cu2+ and colchicine, which improve considerably the overall impression of the work. In general, the paper can be accepted for publication after performing some significant corrections:
1. ESI file is now absent in the submission, so the mentioned PXRD data are uncheckable. Please add ESI to the revised submission.
Response: 1. Thank you for your suggestions, we have added the ESI to the revised submission.
2. PXRD confirmation of the framework stability in H2O is not mentioned. Please add this information to the manuscript or ESI.
Response: 2. Thank you for your suggestions,we have measured the PXRD of the compound after immersing it in distilled water for 24 h, the structure of the compound remained stable, and we put the corresponding graph in the supporting information (Figure. S4).
3. Fe3+, Cu2+, NACs and colchicine have their own strong absorption in UV, so the UV absorption of free moietties in solutions can easily affect the observable concentrational quenching. To confirm the real impact of these species on MOF luminescence, their incorporation needs to be additinally proven by EDX/OES (for metals) or by CHN/IR/NMR/another method (for organics).
Response: 3. Thank you for your suggestions, we have done the UV adsorption of MOF, TNP and colchicine, then we put the graph in Fig. S13, and explained the interaction of them at the end of this manuscript, then we marked it blue.
4. line 11 - the compound is not based on phenanthroline, despite using this reagent in the synthesis. Please correct this sentence.
Response: 4. Thank you for your suggestions, we have used phenanthroline as the starting materials, then we synthesized this compound, otherwise, we could not synthesize the compounds without phenanthroline, and if we add double amount phenanthroline, it could be another known structure.
5. At least some assumptions concerning the mechanism of metal ion / NAC / colchicine impact on the MOF luminescence should be suggested in the main text.
Response: 5. Thank you for your suggestions, we cited reference 12, and explained the impact of MOF with metal ions, TNP and colchicine at the end of this manuscript, then we marked it blue.
6. Of course most chemists know main applications of colchicine, but it would be fair to add a brief description of colchicine importance into the introduction part.
Response: 6. Thank you for your suggestions, we have added the importance of colchicine in the introduction part and marked it in blue.
Reviewer 3 Report
Comments on the work done by Zhou et al entitled “A water-stable Zn-MOF used as multiresponsive luminescent probe for sensing Fe3+/Cu2+, TNP and colchicine in aqueous medium” submitted to the Materials Journal
Manuscript ID: materials-1929115
In this paper, the authors have constructed a water-stable Zn-MOF and they have tried to characterize its topology. By this paper, the authors showed that this compound can be used as a versatile chemosensor. Although this paper does not present a novel hypothesis about a related research field, it presents experimental fresh findings over the manuscript. I think it is a positive contribution to literature as material synthesis and its test with modern technology. I believe that these types of useful material studies could be beneficial for related literature readers. I did not scan tests about plagiarism using any commercial software. So, I assume that it is passed from it before starting the review process by the journal authorities.
Consequently, according to my opinion, this work could be a good candidate for the Materials Journal obeying the other reviewers’ decisions and adhering to the ultimate opinion of the Editor.
General comments:
- The topic is current and of interest to related literature.
- The idea of the study is worth to work and is well-presented in the text.
- The claimed study is logical, synthesis of the compound and physical measurements were successfully done by the authors.
- The references are complete, appropriate, and up to date.
- Tables and figures are appropriate and well-designed.
- I can suggest that the authors should give uncertainty values when giving the result. For example, emission wavelengths
- Using symbols alongside colours when presenting figures can be beneficial for readers with colourless printouts.
Specific comments:
- the authors have efficiently summarized the previous studies on the field and the importance of the study. However, their aim and the reason for the usage of the compound are relatively weak in the introduction part. Therefore, I can suggest more emphasizing their material, at least one more paragraph
-Although reporting of the results and figures is adequate, the discussion of the results needs more scientifically details and comparing their findings with related literature.
- I believe that relative words such as “successive, excellent sensitivity, etc.” are not for scientific presentation. I think that it would be better to present consistency values together with error calculations.
Author Response
Manuscript ID: materials-1929115
In this paper, the authors have constructed a water-stable Zn-MOF and they have tried to characterize its topology. By this paper, the authors showed that this compound can be used as a versatile chemosensor. Although this paper does not present a novel hypothesis about a related research field, it presents experimental fresh findings over the manuscript. I think it is a positive contribution to literature as material synthesis and its test with modern technology. I believe that these types of useful material studies could be beneficial for related literature readers. I did not scan tests about plagiarism using any commercial software. So, I assume that it is passed from it before starting the review process by the journal authorities.
Consequently, according to my opinion, this work could be a good candidate for the Materials Journal obeying the other reviewers’ decisions and adhering to the ultimate opinion of the Editor.
General comments:
- The topic is current and of interest to related literature.
- The idea of the study is worth to work and is well-presented in the text.
- The claimed study is logical, synthesis of the compound and physical measurements were successfully done by the authors.
- The references are complete, appropriate, and up to date.
- Tables and figures are appropriate and well-designed.
1. I can suggest that the authors should give uncertainty values when giving the result. For example, emission wavelengths
Rsponse 1: Thank you for your suggestions, we have added the excitation, emission wavelengths to Fig. 6-7, Fig.9-10.
2. Using symbols alongside colours when presenting figures can be beneficial for readers with colourless printouts.
Response 2: Thank you for your suggestions, we have changed colours of the symbols in the manuscript, and then marked them in blue.
Specific comments:
3. The authors have efficiently summarized the previous studies on the field and the importance of the study. However, their aim and the reason for the usage of the compound are relatively weak in the introduction part. Therefore, I can suggest more emphasizing their material, at least one more paragraph.
Response 3: Thank you for your suggestions, we have added one sentence concerning about the usage of the compound in the introduction part, and marked it in blue.
4. Although reporting of the results and figures is adequate, the discussion of the results needs more scientifically details and comparing their findings with related literature.
Response 4: Thank you for your suggestions, we have added some references in the manuscript, such as references 12,13,14.
5. I believe that relative words such as “successive, excellent sensitivity, etc.” are not for scientific presentation. I think that it would be better to present consistency values together with error calculations.
Response 5: Thank you for your suggestions, we have deleted the words of “successive, excellent sensitivity”, and we have also added the consistency values together with error calculations in the supporting information part, they are Fig.S6-S7, Fig.S10 and Fig.S12.
Round 2
Reviewer 2 Report
The paper can now be accepted for publication. Please change Adsorption to Absorption in Figure S13 at the final proofreading stage.
Author Response
The paper can now be accepted for publication. Please change Adsorption to Absorption in Figure S13 at the final proofreading stage.
Response: Thank you for your suggestions, we have changed “Adsorption” to “ Absorption” in Figure S13, and marked it in blue in the supporting information.